# Maternal Obesity Modifies the Impact of Active SARS-CoV-2 Infection on Placental Pathology

**DOI:** 10.3390/v17071013

**Published:** 2025-07-18

**Authors:** Francisca Carmo, Carla Ramalho, Susana Guimarães, Fátima Martel

**Affiliations:** 1Unit of Biochemistry, Department of Biomedicine, Faculty of Medicine, University of Porto, 4200-319 Porto, Portugal; up201606420@up.pt; 2Instituto de Investigação e Inovação em Saúde (i3S), University of Porto, 4200-135 Porto, Portugal; 3Department of Obstetrics, Unidade Local de Saúde de São João, 4200-319 Porto, Portugal; carlaramalho@med.up.pt; 4Department of Obstetrics-Gynecology and Pediatrics, Faculty of Medicine, University of Porto, 4200-319 Porto, Portugal; 5Department of Pathology, Unidade Local de Saúde de São João, 4200-319 Porto, Portugal; susanamelo@med.up.pt; 6Department of Pathology, Faculty of Medicine, University of Porto, 4200-319 Porto, Portugal

**Keywords:** placenta, histology, SARS-CoV-2, maternal COVID-19, obesity

## Abstract

Background: Obesity during pregnancy is associated with an elevated risk of severe COVID-19, including higher rates of maternal complications, intensive care admission, and adverse neonatal outcomes. The impact of combination of SARS-CoV-2 infection and maternal obesity in placental pathology has not been properly investigated. Aim: To compare the histopathological changes in the placenta induced by active SARS-CoV-2 infection in obese and non-obese patients. Methods: This retrospective cohort study included human placentas from non-obese women and pre-gestationally obese women with active SARS-CoV-2 infection (SARS and OB+SARS, respectively), and placentas from non-obese women and pre-gestationally obese women without SARS-CoV-2 infection (control and OB, collected in the post- and pre-pandemic periods, respectively). Results: A higher (50%) occurrence of ischemic injury and subchorionic fibrin deposits and a 15× higher risk of occurrence of these lesions were found in the OB+SARS group, in relation to control. In contrast, a 10% lower risk of developing chorangiosis in the OB+SARS group than the OB group was observed. Conclusions: An increased risk of lesions related to both maternal and fetal malperfusion and ischemic injury and a lower risk for chorangiosis exist in placentas from obese women affected by SARS-CoV-2 infection. Importantly, these differences were not observed in placentas from non-obese women.

## 1. Introduction

Emerging in December 2019 in Wuhan, China, and being declared a pandemic in March 2020 [1], the novel coronavirus disease (COVID-19) has affected millions of people worldwide due to its high contagiousness. Severe acute respiratory syndrome coronavirus 2 (SARS-CoV-2) from the family Coronaviridae, genus Betacoronavirus, and species, Severe acute respiratory syndrome-related coronavirus, is the coronavirus associated with COVID-19 that produces the typical symptoms of a viral pneumonia, such as fever, shortness of breath, malaise, and dry cough in humans [2,3].

Compared to the general population, pregnant women show higher mortality rates and complications associated with viral infections [4]. Consistent with this, pregnant women are at increased risk of developing severe forms of COVID-19, with higher rates of hospitalization and intensive care unit (ICU) admission compared to non-pregnant women in the same age group. Additionally, COVID-19 during pregnancy increases the risk of pregnancy complications such as preeclampsia, miscarriage, preterm birth, and stillbirth, and is associated with higher rates of cesarian delivery mode and increased maternal and neonatal morbidity and mortality [5]. 

The placenta acts as the interface between the mother and the fetus during pregnancy. It allows for the exchange of nutrients and gases and for the removal of fetal waste products, having a crucial role in supporting fetal development and metabolism [6,7]. Histopathologic examination of placental tissue can provide significant information regarding the health of both mother and fetus. Several viral infections in pregnancy are associated with aberrant placental findings, including diffuse high-grade villitis, the presence of plasma cells with associated enlargement of villi, and hemosiderin deposition in the villi [8,9].

Although placental infection and vertical transmission of the SARS-CoV-2 virus to the fetus has a low probability [10], analysis of placental sections from COVID-19 mothers usually show increased perivillous fibrin deposition, chronic and acute inflammatory pathology, including chronic histiocytic intervillositis, and maternal and/or fetal vascular malperfusion, including intervillous thrombosis and decidual arteriopathy [11,12,13]. Of note, although extensive research has investigated the effect of SARS-CoV-2 infection during pregnancy at the placental level, a clear consensus as to the relevance or even the higher occurrence rates of these lesions in COVID-19 mothers, when compared to the general population, does not exist [11,14].

Due to the current obesity epidemic, there is a high prevalence of obesity in women of child-bearing age [15]. Maternal obesity is linked to greater maternal and fetal morbidity and to the development of hypertensive disorders in pregnancy, gestational diabetes, fetal macrosomia, congenital defects, and stillbirth [16,17]. When affected by COVID-19, obese pregnant women are at a greater risk of developing more severe symptoms of this disease, thromboembolism, and other infections, and have a greater risk of ICU admission, non-invasive and invasive oxygen therapy, and even death [18,19,20]. The increased vulnerability to the disease appears to be linked to the obesity-associated reduction in respiratory reserves and elevation of circulating inflammatory factor levels [19]. The coexistence of pregnancy—a pro-inflammatory and immuno-tolerant state [21]—further exacerbates these phenomena.

At the placental level, maternal obesity is associated with the occurrence of several lesions, such as maternal [22,23] and fetal [23] vascular malperfusion, decidual arteriopathy [23] and vasculopathy [24], delayed villous maturation [25], placental infarcts [23], increased fibrin deposition and increased syncytial knots [26], and even chronic villitis [24] and vasculitis [27]. Importantly, the placenta plays a central role in the programming events resulting from maternal obesity that lead to the higher risk of the development of obesity/metabolic syndrome in the offspring [16,17,28,29].

There is growing evidence that the placenta is particularly susceptible to the dual impact of maternal obesity and SARS-CoV-2 infection [30], but the possible impact that the combination of SARS-CoV-2 infection and maternal obesity can have in placental pathology has not been properly investigated. So, the aim of this study was to compare the placental histopathology changes induced by an active SARS-CoV-2 infection in obese and non-obese mothers.

## 2. Methods

This was a retrospective cohort study approved by The Research Ethics Committee of Centro Hospitalar São João (CHUSJ), Portugal (n° 125/22), under the project entitled ‘A influência da infeção por COVID-19 durante a gravidez na placenta.’

### 2.1. Case Selection

This study included placentas from deliveries in ULS São João between December 2017 and July 2023, selected to form four study groups. The inclusion criteria for this study were: RT-PCR test positive for SARS-CoV-2 at the time of admission for delivery, live fetus, placental pathological report available, and no COVID-19 vaccination. The exclusion criteria for this study were multiple pregnancies, stillbirth, fetal death, tobacco and drug use during pregnancy, maternal comorbidities such as HIV infection, cholestasis, systemic lupus erythematosus, rheumatoid arthritis, Crohn disease and renal disease, pregnancy-associated diseases such as preeclampsia and intrauterine growth restriction (IUGR), and recovered SARS-CoV-2 infection at the time of labor. SARS-CoV-2 testing was performed using standard protocols. Nasopharyngeal and oropharyngeal swabs were collected from expectant mothers at the time of admission to delivery and used for RT-PCR.

The control group (*n* = 17) included placentas collected from healthy mothers (body mass index [BMI] < 30) with singleton pregnancies, recruited for this study between June 2022 and July 2023 (post-pandemic). The obese (OB) group (*n* = 14) corresponded to a convenience sample, obtained from pre-pandemic historical records of the Pathology Department of ULS São João (from December 2017 to April 2019), of placentas obtained from pregnant women with pre-gestational obesity (BMI ≥ 30). These placentas had an indication for pathologic evaluation that did not significantly interfere with placental pathology (i.e., minor fetal abnormalities such as isolated cleft palate). The SARS group (*n* = 19) included placentas from healthy mothers (BMI < 30) with singleton pregnancies, presenting with active SARS-CoV-2 infection at the time of delivery that were recruited for this study between April 2020 and May 2022. The OB+SARS group (*n* = 12) included placentas from obese mothers (pre-gestational BMI ≥ 30) with singleton pregnancies with active SARS-CoV-2 infection at the time of delivery that were recruited for this study between April 2020 and May 2022. Both SARS-CoV-2 affected groups were composed of patients who had not been vaccinated for COVID-19 at the time of infection and delivery. As the exclusion of cases presenting with gestational diabetes (GDM) from the OB and OB+SARS groups would lead to very small study groups, we decided to include GDM in similar percentages in both groups (5/14 cases in the OB group and 4/12 cases in the OB+SARS group). Because GDM has a known impact on placental pathology, we decided to also include patients with GDM in the control (4/17 cases) and SARS (3/19 cases) groups. Informed consent was obtained from all subjects involved in the study.

### 2.2. Data Collection

The following clinicopathological patient data were collected: maternal age, parity, number of gestations, body mass index (BMI), gestational age at delivery, mode of delivery, newborn weight, APGAR score at 5 min, presence of maternal GDM and HT, history of SARS-CoV-2 infection, gestational age at COVID-19 diagnosis, classification of COVID-19 symptoms, ICU and/or neonatal intensive care unit (NICU) admission, and need for invasive ventilation. The classification of COVID-19 symptoms was made according to the NIH symptom severity criteria [31].

All placentas were examined by experienced pathologists who were aware of group status. The macroscopic examination included the documentation of placental weight and size, umbilical cord size, number of vessels and coiling, and presence of chorionic, basal, or parenchymal abnormalities. The placental weight percentile and feto-placental ratio percentile were calculated and categorized as diminished if the placenta or ratio was below the 10th percentile, normal if the placenta or ratio was equal to or higher than the 10th percentile and lower than 90th percentile, and increased if the ratio was equal to or higher than the 90th percentile. Samples were taken from fetal and placental ends of the umbilical cord, one membrane roll, three normal parenchymal sections (full thickness), and parenchymal lesions. Formalin-fixed, paraffin-embedded tissues were sectioned in 3 µm thickness and stained with H&E for histopathological evaluation, which was made according to the Amsterdam criteria [32]. Findings were categorized into seven major categories: gross findings and anatomic alterations, features of maternal vascular malperfusion (MVM) and fetal vascular malperfusion (FVM), inflammatory pathology (villitis, chronic histiocytic intervillositis, deciduitis), and acute infectious pathology (chorioamnionitis).

### 2.3. Statistical Analysis

The statistical analysis was performed using IBM SPSS Statistics for Windows (Version 19.0) (IBM Corp, Armonk, NY, USA). Continuous variables are presented with mean ± SD, or median and IQR when variables followed non-normal distribution. Categorical variables are presented with frequencies and percentages. For associations between categorical variables, the chi-square test or, when applicable, Fisher’s exact test was used. For continuous variables, the *t*-test or, when applicable, the Mann–Whitney U test was used for comparisons between two groups, and for multiple comparisons, ANOVA or, when applicable, the Kruskal–Wallis test was used. A *p*-value of 0.05 was used for statistical significance.

## 3. Results and Discussion

### 3.1. Maternal and Fetal Clinicopathological Characteristics

The clinicopathological data of the four study groups are shown in Table 1. The mean maternal age was not significantly different between the groups, the same applying to the parity and number of gestations of the participants. As anticipated, the body mass index (BMI) was higher in the OB and OB+SARS groups compared to the non-obese groups. Furthermore, the mean BMI in both non-obese groups (control vs. SARS groups) and in both obese groups (OB vs. OB+SARS groups) were not statistically different.

As to the mean gestational age at delivery, it was only significantly lower in the OB+SARS group compared to the SARS group. This may be due to the inclusion of two pregnancies in which there was preterm induction of labor in the OB+SARS group (one at 34 weeks and the other at 29 weeks of pregnancy), related to a decrease in fetal movements and the need for invasive maternal ventilation due to COVID-19 infection. Of note, despite the preterm birth in the OB+SARS group, placental and neonatal weight were not significantly affected (Table 1 and Table 2).

In relation to fetal status at the time of admission, in the SARS group, there was a case with a non-reassuring fetal status and one case of fetal macrosomia. In the OB group, there were two cases of fetal macrosomia and three cases with non-reassuring fetal status, in which the newborns had to be later admitted to the NICU; in one of these cases, the Apgar score of the newborn was low (4 at 1′, 6 at 5′ and 7 at 10′ after birth). In the SARS+OB group, there were two cases of fetal macrosomia and two cases in which the newborns were admitted to NICU.

Cesarian delivery tended to be more frequent in both the OB and the OB+SARS groups, which agrees with previous works describing a higher risk of cesarian section for obese patients [33,34,35] and for patients with active SARS-CoV-2 infection [36,37,38,39,40]. However, in the present study, the differences were not statistically significant.

The characteristics of the newborns (Apgar score at 5 min and weight) were also similar in the four groups.

### 3.2. Severity of SARS-CoV-2 Infection

When comparing the characteristics of COVID-19 in the two SARS-CoV-2 infected groups (SARS and OB+SARS) (Table 3), more symptomatic infections were observed in the OB+SARS group. Moreover, moderate to severe symptoms were exclusive to this group, as the SARS group consisted only of asymptomatic and mild infection cases. This result is in accordance with previous works that concluded that the risk of developing more severe COVID-19 is higher in obese patients [41,42]. In the two groups affected by SARS-CoV-2 infection, the recorded symptoms were, in order of frequency: dry cough and anosmia, pneumonia, fever, productive cough, myalgia, dyspnea, fatigue, and dysgeusia. The two severe cases belonging to the OB+SARS group were admitted to the ICU, developed acute respiratory distress syndrome (ARDS), and needed invasive oxygen therapy. Of note, there were several cases in both groups with missing information regarding the presence/absence of symptomatic SARS-CoV-2 infection and the severity of symptoms.

Finally, most likely because these two groups (SARS and OB+SARS) included only patients with active SARS-CoV-2 infection at the time of delivery, the gestational age at diagnosis of SARS-CoV-2 infection was similar in these two groups.

### 3.3. Macroscopic Evaluation of the Placenta

As shown in Table 2, none of the parameters evaluated (placental weight, placental size, fetal–placental weight ratio percentile, and umbilical cord coiling index) were significantly different among the four groups. This agrees with previous studies that found no association between SARS-CoV-2 infection and placental macroscopic findings in either non-obese [43,44] or obese patients [30]. However, obesity is commonly associated with large-for-gestational-age placentas and a reduced feto-placental weight ratio [45,46]. 

### 3.4. Histological Evaluation of the Placenta

Villitis. Reports of villitis are common in previous studies of SARS-CoV-2 [36,47,48] and obese pregnancies [49,50]. Accordingly, although not significantly different, we observed a higher percentage of placentas with villitis in both the OB and the OB+SARS groups (Table 4).

Intervillositis. The occurrence of chronic histiocytic intervillositis was very low in our study and not significantly different among groups. A previous study with a SARS-CoV-2 group containing mainly active infections at delivery (70% of the cohort) referred to a higher rate of chronic histiocytic intervillositis in this group than in the control group [51]. This difference was not verified in our study (Table 4) or in other studies only including active infections [38,52] (the latter with a SARS-CoV-2 + group comprised of mainly obese patients with mean BMI 33.4 kg/m^2^ and standard deviation 7.7 kg/m^2^). 

Deciduitis. No significant differences were found in relation to chronic deciduitis among the four groups; although, in terms of the percentage of cases, they were higher in both SARS-CoV-2 affected groups (Table 4). In contrast to our findings, an increased occurrence of chronic deciduitis in cases with active SARS-CoV-2 infection, when compared to control, was previously described [53], and an increased occurrence of chronic deciduitis cases in a mainly obese population (mean BMI 31.6 kg/m^2^ and 12.5 kg/m^2^ interquartile range) with active SARS-CoV-2 infection, when compared to the global percentage of occurrence of this lesion, was described [37]. On the other hand, another study [54] found that chronic deciduitis was not associated with obese pregnancies. 

Chorioamnionitis. Acute chorioamnionitis was observed more frequently in the SARS group, although with no statistical significance (Table 4). Although not directly associated with SARS-CoV-2 (as it is of bacterial etiology), these findings have been found to be more prevalent in SARS-CoV-2-infected [55] and obese mothers [46,49], even though this is not universally coherent in SARS-CoV-2 infection [39,56,57] or in SARS-CoV-2 infection associated with maternal obesity [38].

Villous maturation. Although no differences were observed in the distribution of villous maturation types between the groups, a high proportion of placentas with delayed villous maturation was present across all groups (Table 4). Delayed villous maturation was also previously described in obese pregnancies [25] and is a finding commonly associated with GDM [58,59], a pathology present in a fraction of cases in all our groups. In contrast, an increase in the occurrence of accelerated villous maturation was previously described in obese pregnancies [49], in active SARS-CoV-2-associated pregnancies [60,61], and in mainly obese (mean BMI 32.6 kg/m^2^ and range 19.8–51.4 kg/m^2^) and SARS-CoV-2-associated pregnancies [62].

Maternal vascular malperfusion (MVM). MVM was significantly different in the four groups. When individually comparing the groups, MVM was more common in the two SARS-CoV-2-infected groups (SARS and OB+SARS groups), with a similar percentage of occurrence in these two groups, than in the control and OB groups. Because the percentages of cases with MVM in the control and OB groups (5.9% and 0%, respectively) tended to be lower than those in the SARS and OB+SARS groups (31.6% and 33.3%) (although the difference was only significant when comparing OB+SARS vs. OB), we suggest that SARS-CoV-2 active infection may lead to increased features of MVM. It is noteworthy that previous studies did not clearly establish a relationship between MVM and SARS-CoV-2 infection. Indeed, in some studies, active SARS-CoV-2 infection was associated with an increase in the occurrence of [53,57,61] or risk for developing MVM [63], but other studies found no difference [38,52,64,65,66,67]. In obese patients, this pattern of lesions was also found to be more prevalent than in non-obese patients [44,45,63]. Finally, Brien et al. [62] also found a higher prevalence of MVM in a SARS-CoV-2 affected and mainly obese cohort, when compared to a mainly obese cohort, but the same results were not obtained by Lu-Culligan [38] in similar cohorts.

Placental infarction. Although no significant differences were found between the four groups, placental infarction was only observed in the two SARS-CoV-2 affected groups (Table 4). A higher occurrence of placental infarction was previously reported in cohorts with active SARS-CoV-2 infection compared to controls [36,40,57]. Moreover, Leon-Garcia et al. [24], Åmark et al. [27], and Beneventi et al. [49] did not find any association between placental infarction and maternal obesity, but Huang et al. [68] did find an increased occurrence in obese placentas compared to controls.

Distal villous hypoplasia, villous agglutination, and increased syncytial knotting. These parameters were similar in the four groups of placentas, although some tendencies were noted (Table 4). In contrast to our observation, Şahin et al. [39] found distal villous hypoplasia to be more prevalent in the SARS group, but Beneventi et al. [49] did not find any association between this lesion and obese pregnancies. In relation to villous agglutination, opposite effects were previously found: It occurred more frequently in control groups than in active SARS-CoV-2 groups [69], but the opposite result was found by Singh et al. [65]. Finally, increased syncytial knotting has been previously associated with active SARS-CoV-2 [36,65,70] and with maternal obesity [68].

Hypoxic-ischemic lesions. In comparison to the control group, these lesions were more common in the SARS and OB+SARS groups (Table 4). Additionally, the risk tended to be higher in the SARS group (Table 5) and was 15 times higher in the OB+SARS group, in relation to the control group (Table 5). This shows that the occurrence of this type of injury can be linked to SARS-CoV-2 infection. Accordingly, a higher occurrence of ischemic injury was linked to SARS-CoV-2 infection in a previous study [36]. This pattern of injury is associated with impaired placental blood flow, infarctions, and oxidative damage. When present in significant amounts, these lesions can contribute to intrauterine growth restriction (IUGR), abnormal neurodevelopment, and long-term health consequences [71,72,73].

Fetal vascular malperfusion (FVM). FVM occurred in low proportions in all the groups, and no statistically significant differences were observed (Table 4). This contradicts previous research with active SARS-CoV-2 [53,61,64,74], obese [23,49], and mainly obese with active SARS-CoV-2 [75] pregnancies. The related finding of avascular villi was also not very prevalent in our study (Table 4).

Subchorionic fibrin deposits. When evaluated both macro- and microscopically, subchorionic fibrin deposits were significantly more frequent in the OB+SARS group than in both the control and OB groups (Table 4). Also, the risk of developing this lesion tended to be higher in the SARS group than in the control group (Table 5). Thus, this lesion seems to be associated with SARS-CoV-2 infection, as previously shown by Carbonnel et al. [57], but additionally, we found that it is aggravated by the combination of maternal obesity and infection. This observation is in discordance with a previous study that did not verify any differences between two mainly obese groups: one with active COVID-19 and one without (mean BMI 32.5 kg/m^2^ and standard deviation 5.5 kg/m^2^, and mean BMI 32.1 kg/m^2^ and standard deviation 6.0 kg/m^2^, respectively) [75]. However, similar to our observation, this lesion was previously shown to be unchanged in obese pregnancies [68].

Perivillous fibrin deposition. Perivillous fibrin deposition tended to be more common in the OB+SARS group compared to the others, but this result did not reach statistical significance (Table 4). Some placentas had a diagnosis of massive ‘like’ perivillous fibrin deposition: two placentas in the SARS group with 25% and 5% of parenchyma affected, one in the OB group with 5% of parenchyma affected, and one in the OB+SARS group with 15% of parenchyma affected. Two more diagnoses of massive ‘like’ perivillous fibrin deposition were made, one in the SARS group and one in the OB group, in which the percentage of affected parenchyma was not specified. The deposition of perivillous fibrin, sometimes in a massive manner, is one of the more frequent placental lesions in active SARS-CoV-2 placentas [53,61,65,70], and has also been reported to have a higher prevalence in placentas from a cohort of mainly obese women with active [37,38] or both active and recovered [62] SARS-CoV-2 infections, when compared to the respective controls.

Perivillous thrombus. A tendency for a higher proportion of perivillous thrombus in the control and OB+SARS groups than in the SARS and OB groups was found, but this difference was not significant (Table 4). Previous studies showed no association between this lesion and active SARS-CoV-2 infection [52,66], but He et al. [46] found it to be increased in obese placentas compared to controls.

Intramural fibrin deposition. The occurrence of intramural fibrin deposition was statistically different between all the groups (Table 4). Because we found a higher occurrence of these lesions in the OB+SARS groups than in the SARS group, and because no other differences between these groups were found, the higher risk of occurrence is probably due to the coexistence of SARS-CoV-2 infection with maternal obesity (Table 5). Previously, a higher prevalence of intramural thrombi in SARS-CoV-2 pregnancies than in control pregnancies was observed in both non-obese [61] and mainly obese [75] cohorts. Finally, the same type of lesion has also been described more frequently in obese-derived placentas than in non-obese-derived placentas [49].

Fibrin deposition. We also verified that placentas belonging to all four groups possessed a similarly high prevalence of fibrin deposition of any feature (Table 4).

Chorangiosis. Lastly, differences between the groups exist in relation to the occurrence of chorangiosis (Table 4). Interestingly, a protective effect of active SARS-CoV-2 infection in relation to the occurrence of chorangiosis appears to exist for the obese patients, because the occurrence and risk of developing chorangiosis was lower in the OB+SARS than in the OB group (Table 5). The observation of a tendency of a higher risk of developing chorangiosis in the OB group relative to the control group is in accordance with a previous report [49]. As to SARS-CoV-2, Şahin et al. [39] verified that in the control group, chorangiosis was more common, and Debelenko et al. [52] noted a tendency for a higher occurrence of this lesion in the control group than in active SARS-CoV-2 affected pregnancies.

Placentitis. Contrary to previous studies, we did not verify any signs of the phenomenon described as SARS-CoV-2 placentitis [76,77,78]. We hypothesize that the reason for this difference is related to the fact that, even though we included severe and moderate cases of COVID-19, we only included active infections at birth, so perhaps there was not enough time for this deleterious immunologic reaction to develop.

Our work shows several differences from several previous studies that found no or very few differences between active SARS-CoV-2 affected and non-affected placentas [56,57,64,66,67,69,79]. These disparities could arise from a lack of uniformity in group composition in many of the previous studies. Indeed, several previous works included several maternal comorbidities and pregnancy complications that can act as confounders, which were either paired/included in similar percentages in the different groups (e.g., [39,57,64,69]) or non-paired (e.g., [38,52,67]). Because we were aware of this problem, our groups comprised mainly healthy pregnancies in the control and SARS groups, with the inclusion of only GDM in both groups, to control for pathognomonic lesions of this complication. Furthermore, previous studies evaluating the effect of pre-pregnancy obesity in SARS-CoV-2 related placental pathology did not include a cohort composed only of obese women [30]. As to the present study, its main limitations are the small size of our study groups, the fact that this in an exploratory analysis that did not apply methods to reduce overestimation of significances, the fact that the pathologist was aware of group status, and, in several cases, the missing data. 

In conclusion, we found an increased risk for the development of several placental lesions related to both maternal and fetal vascular malperfusion and ischemic-related injury in placentas affected by SARS-CoV-2, and an increased risk for the development of such injuries when SARS-CoV-2 infection co-occurs with maternal obesity. Furthermore, a lower risk for chorangiosis in placentas from SARS-CoV-2 infected obese women, but not non-obese women, was found. Further studies with larger sample sizes and coherent group construction criteria are needed to further establish the link between these two diseases and how their interaction contributed to the generally worse outcomes observed during the COVID-19 pandemic. Once this relationship is established, it will also be possible to infer for a possible programming effect of obesity associated with SARS-CoV-2 infection during pregnancy.

## Figures and Tables

**Table 1 viruses-17-01013-t001:** General characteristics of the different groups of patients.

		Total	Control	SARS	OB	OB+SARS	4 Groups	Control vs. SARS	Control vs. OB	Control vs. OB+SARS	OB+SARS vs. SARS	OB+SARS vs. OB
Variable		*n* = 62, *n* (%)	*n* = 17, *n* (%)	*n* = 19, *n* (%)	*n* = 14, *n* (%)	*n* = 12, *n* (%)						
Maternal age (y), Mean ± SD		31.2 ± 0.7	31.4 ± 1.2	32.4 ± 1.4	29.7 ± 1.0	30.7 ± 1.8						
Parity (*n*), Median (IQR)		1 (1)	0 (1)	1 (1)	0.5 (1)	0.5 (1.75)						
Gestations (*n*), Median (IQR)		2 (2)	1 (1)	2 (2)	2 (1.25)	2 (2)						
BMI (kg/m^2^), Mean ± SD		28.3 ± 0.8	23.8 ± 0.6	23.7 ± 0.4	35.7 ± 1.9	33.4 ± 0.7	***		***	***	***	
GDM, *n* (%)		16 (25.8%)	4 (23.5%)	3 (15.8%)	5 (35.7%)	4 (33.3%)						
HT, *n* (%)	Gestational	2 (3.2%)	0	0	1 (7.1%)	1 (8.3%)						
	Chronic	1 (1.6%)	0	0	1 (7.1%)	0						
Gestational Age (wk), Mean ± SD		39.1 ± 0.2	39.0 ± 0.2	39.8 ± 0.2	39.2 ± 0.2	37.8 ± 0.9	*				**	
Type of delivery, (*n*)	Vaginal	28 (45.9%)	11 (64.7%)	11 (57.9%)	2 (14.3%)	4 (33.3%)						
	Cesarian	33 (54.1%)	6 (35.3%)	6 (42.1%)	12 (85.7%)	8 (66.7%)						
NB weight (g), Mean ± SD		3383 ± 79	3337 ± 86	3281 ± 99	3310 ± 204	3383 ± 79						
Apgar score (*n*), Mean (IQR)	5′	10 (1)	10 (0)	10 (0)	10 (1)	9.5 (1)						

Statistically significant: *—*p* < 0.05; **—*p* < 0.01; ***—*p* < 0.001; SD, standard deviation; IQR, interquartile range; BMI, body mass index; GDM, gestational diabetes mellitus; HT, hypertension; NB, newborn.

**Table 2 viruses-17-01013-t002:** Comparison of placental macroscopic evaluations between groups.

	Total	Control	SARS	OB	OB+SARS	4 Groups
Variable	*n* = 62, *n* (%)	*n* = 17, *n* (%)	*n* = 19, *n* (%)	*n* = 14, *n* (%)	*n* = 12, *n* (%)	
Placental weight (g), mean ± SD		461.5 ± 11.6	474 ± 21.3	447.7 ± 16.8	490.1 ± 26.6	
Placental size (cm), mean ± SD		18.96 ± 0.2	18.66 ± 0.5	19.96 ± 0.4	19.37 ± 0.7	
Placental size percentile, *n* (%)	Very diminished	3 (4.8)	0	2 (12.5)	0	
	Diminished	3 (4.8)	1 (7.1)	0	1 (8.3)	
	Normal	44 (71)	13 (92.9)	14 (87.5)	11 (91.7)	
	N/A	12 (19.4)	3 (17.6)	3 (15.8)	2 (14.3)	
Fetal–placental weight ratio, *n* (%)	Very diminished	2 (3.2)	1 (7.1)	0	1 (8.3)	
	Diminished	1 (1.6)	0	0	1 (8.3)	
	Normal	37 (59.7)	7 (50)	14 (87.5)	9 (75)	
	Increased	1 (1.6)	1 (7.1)	0	0	
	Very increased	9 (14.5)	5 (35.7)	2 (12.5)	1 (8.3)	
	N/A	12 (19.4)	3 (17.6)	3 (15.8)	2 (14.3)	
Cord coiling, *n* (%)	Undercoiled	3 (5.5)	1 (7.1)	2 (10.5)	0	
	Normal	33 (60)	10 (71.4)	12 (63.2)	5 (50)	
	Hypercoiled	19 (34.5)	3 (21.4)	5 (26.3)	5 (50)	
	N/A	7 (11.3)	3 (17.6)	0	4 (6.4)	

N/A, non-available information. Shown are valid percentages. No differences were verified between the four groups.

**Table 3 viruses-17-01013-t003:** Characteristics of SARS-CoV-2 infection in the two SARS groups.

Groups		Total	SARS	OB+SARS	
Variable		*n* = 31, *n* (%)	*n* = 19, *n* (%)	*n* = 12, *n* (%)	
Symptomatic infection, *n* (%)	Yes	10 (40%)	4 (23.5%)	6 (75%)	*
	N/A	6/19.3%	2/10.5%	4/33.3%	
SARS-CoV-2 infection severity, *n* (%)	Non-symptomatic	15 (60%)	13 (76.5%)	2 (25%)	*
	Mild	7 (28%)	4 (23.5%)	3 (37.5%)	
	Moderate	1 (4%)	0	1 (12.5%)	
	Severe	2 (8%)	0	2 (25%)	
Time of SARS-CoV-2 infection (gestational age week), median (IQR)			39 (1)	38 (4.75)	

Statistically significant: *—*p* < 0.05; N/A, non-available information. Shown are valid percentages.

**Table 4 viruses-17-01013-t004:** Comparison of histopathological findings between groups.

		Total	Control	SARS	OB	OB+SARS	4 Groups	Control vs. SARS	Control vs. OB	Control vs. OB+SARS	OB+SARS vs. SARS	OB+SARS vs. OB
Variable		*n* = 62, *n* (%)	*n* = 17, *n* (%)	*n* = 19, *n* (%)	*n* = 14, *n* (%)	*n* = 12, *n* (%)						
Villitis, *n* (%)		13 (21)	2 (11.8)	3 (15.8)	4 (28.6)	4 (33.3)						
Chronic histiocytic intervillositis	Grade 1	3	1 (5.9)	1 (5.3)	1 (7.1)	0						
Deciduitis, *n* (%)	Chronic	19 (30.6)	3 (17,6)	7 (36.8)	3 (21.4)	6 (50)						
Acute Chorioamnionitis, *n* (%)		23 (37.1)	4 (23.5)	9 (47.4)	6 (42.9)	4 (33.3)						
Chorioamnionitis Maternal response	Stage 1 Grade 1	7	1	5	1	0						
	Stage 2 Grade 1	5	1	1	1	2						
	Stage 2 Grade 2	7	1	4	0	2						
Chorioamnionitis Fetal response	Stage 2 Grade 1	7	1	4	1	1						
	Stage 2 Grade 2	3	0	1	1	1						
	Stage 3 Grade 2	1	0	0	1	0						
Villous Maturation, *n* (%)	Delayed	28 (45.2)	8 (47.1)	8 (42.1)	7 (50)	5 (41.7)						
	Normal	31 (50)	8 (47.1)	11 (57.9)	7 (50)	5 (41.7)						
	Accelerated	3 (4.8)	1 (5.9)	0	0	2 (16.7)						
MVM, *n* (%)	Formal diagnosis	11 (17.7)	1 (5.9)	6 (31.6)	0	4 (33.3)	*					*
	Any feature	58 (93.5)	16 (94.1)	19 (100)	12 (85.7)	11 (91.7)						
Placental infarcts		4 (6.5)	0	3 (15.8)	0	1 (8.3)						
Distal villous Hypoplasia		14 (22.6)	6 (35.3)	2 (10.5)	4 (28.6)	2 (16.7)						
Increased Syncytial knots		13 (21)	2 (11.8)	3 (15.8)	4 (28.6)	4 (33.3)						
Villous Agglutination		12 (19.4)	4 (23.5)	4 (21.1)	2 (14.3)	2 (16.7)						
Hipoxic ischemic lesions		22 (35.5)	2 (11.8)	8 (42.1)	4 (28.6)	8 (66.7)	*	*		**		
FVM		9 (14.5)	2 (11.8)	3 (15.8)	2 (14.3)	2 (16.7)						
Avascular villi		4 (6.5)	1 (5.9)	2 (10.5)	0	1 (8.3)						
Fibrin deposition	Subchorionic	41 (66.1)	7 (41.2)	14 (73.7)	8 (57.1)	12 (100)	**			***		*
	Perivillous	23 (37.1)	4 (23.5)	6 (31.6)	5 (35.7)	8 (66.7)						
	Massive ‘like’ Perivillous	6 (9.7)	0	3 (15.8)	1 (7.1)	2 (16.7)						
	Pervillous Thrombi	21 (33.9)	7 (41.2)	5 (26.3)	4 (28.6)	5 (41.7)						
	Intramural Thrombi	7 (11.3)	1 (5.9)	0	3 (21.4)	3 (25)	*				*	
	Any feature	55 (88.7)	13 (76.5)	18 (94.7)	12 (85.7)	12 (100)						
Chorangiosis		13 (21)	3 (17.6)	2 (10.5)	7 (50)	1 (8.3)	*					*

Statistically significant: *—*p* < 0.05; **—*p* < 0.01; ***—*p* < 0.001; N/A, non-available information. Shown are valid percentages.

**Table 5 viruses-17-01013-t005:** Odds ratio of histopathological findings among groups.

Variable		Comparison			
			OR	95% CI	*p*-Value ^†^
MVM	Formal diagnosis	Ctrl vs. SARS	7.385	0.786–69.361	0.08
		Ctrl vs. OB	°	°	0.999
		Ctrl vs. OB+SARS	8	0.763	0.083
		OB+SARS vs. SARS	0.923	0.198–4.312	0.919
		OB vs. OB+SARS	°	°	0.998
Hipoxic ischemic lesions		Ctrl vs. SARS	5.455	0.963–30.886	0.055
		Ctrl vs. OB	3	0.459–19.592	0.251
		Ctrl vs. OB+SARS	15	2.239–100.483	0.005 *
		OB+SARS vs. SARS	0.364	0.081–1.641	0.188
		OB vs. OB+SARS	5	0.942–26.530	0.059
Fibrin deposition	Subchorionic	Ctrl vs. SARS	4	0.981–16.311	0.053
		Ctrl vs. OB	1.905	0.454–7.983	0.378
		Ctrl vs. OB+SARS	°	°	°
		OB+SARS vs. SARS	°	°	°
		OB vs. OB+SARS	°	°	°
	Intramural thrombi	Ctrl vs. SARS	°	°	0.0998
		Ctrl vs. OB	4.364	0.4–47.614	0.227
		Ctrl vs. OB+SARS	5.333	0.481–59.144	0.173
		OB+SARS vs. SARS	°	°	0.998
		OB vs. OB+SARS	1.222	0.197–7.594	0.830
Chorangiosis		Ctrl vs. SARS	0.549	0.08–3.760	0.541
		Ctrl vs. OB	4.667	0.916–23.785	0.064
		Ctrl vs. OB+SARS	0.424	0.039–4.662	0.483
		OB+SARS vs. SARS	1.294	0.104–16.043	0.841
		OB vs OB+SARS	0.091	0.009–0.906	0.041 *

Statistically significant: * *p* < 0.05; ^†^ Statistical significance of OR; ° impossible to calculate. The first group in the comparison is the reference group in OR calculation.

## Data Availability

Data are contained within the Appendix A.

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
