# Peer review of "Maternal Obesity Modifies the Impact of Active SARS-CoV-2 Infection on Placental Pathology"

_viruses, 2025, doi:10.3390/v17071013_

Round 1

Reviewer 1 Report

Comments and Suggestions for Authors

I was given the oportunity to review the manuscript on "placental pathology of SARS-CoV-2 affected pregnancies of obese and non-obese patients". The hypothesis is interesting. Obesity is a risk factor for pregnancies, SARS-CoV-2 could be a risk factor for pregnancies, so the combination of both could result in adverse obstetric outcomes.

The authors aknowledge study limitations. Sample size is small and results in small groups (17, 19, 14, and 14 cases per group). Small groups are more susceptible to high variability (a single change in a variable represents a big variation in each group). The authors argue that larger groups would result in statisitcal significance in the differences. This can be true, but small groups are sensitive to biass. Larger sample size could reduce differences based on biasses as well. SARS-CoV-2 positive groups are likely the larger possible, but obese and normal control patients could be increased and the most promising differences should be re-tested.

The authors aknowledge that pathologist (-s) were not blinded.

The authors produce almost a hundred of statistical tests. This results in significant values just by chance. the alpha value (alpha risk) is the risk of taking a false positive as true positive, and it's stated in 5%. This means that just by chance, up to 5 test result positive. Applying Bonferroni correction, none of the statistically significative results would remain.

Author Response

Response to Reviewer 1 Comments

Thank you very much for taking the time to review this manuscript. Please find the detailed responses below and the corresponding revisions/corrections highlighted in the re-submitted file.

1. The English could be improved to more clearly express the research.

Response 1. The manuscript has been reviewed for English improvement.

2. Questions for General Evaluation

Reviewer’s Evaluation

Response and Revisions

Does the introduction provide sufficient background and include all relevant references?

Yes

Is the research design appropriate?

Must be improved

We commented on this issue in response 1 and 2

Are the methods adequately described?

Yes

Are the results clearly presented?

Yes

Are the conclusions supported by the results?

Are all figures and tables clear and well-presented

Must be improved

Yes

We commented on this issue in response 2

3. Point-by-point response to Comments and Suggestions for Authors

Comment 1: The authors acknowledge study limitations. Sample size is small and results in small groups (17, 19, 14, and 14 cases per group). Small groups are more susceptible to high variability (a single change in a variable represents a big variation in each group). The authors argue that larger groups would result in statistical significance in the differences. This can be true, but small groups are sensitive to bias. Larger sample size could reduce differences based on biases as well. SARS-CoV-2 positive groups are likely the larger possible, but obese and normal control patients could be increased and the most promising differences should be re-tested.

Response 1: Thank you for pointing this out. We agree with this comment, although it is currently impossible for us to introduce new participants in this study. We now acknowledge that the statement ‘…we think that some of the non-significant differences that we observed would otherwise become statistically significant with a larger sample’’ is not entirely correct, and so we eliminated it in the revised manuscript (lines 377-379).

Comment 2: The authors produce almost a hundred of statistical tests. This results in significant values just by chance. the alpha value (alpha risk) is the risk of taking a false positive as true positive, and it's stated in 5%. This means that just by chance, up to 5 test result positive. Applying Bonferroni correction, none of the statistically significative results would remain.

Response 2: Although your comment is accurate, we would like to point out several issues. First, this is an exploratory study, that focuses on discovering what possible placental lesions may be caused by this infection and which are aggravated in the presence of maternal obesity. So, we believe that, by narrowing the number of statistical tests we perform (in order to increase the value of the acceptable α), we would be losing a lot of potentially important information. Second, it is usual to use α= 0.05 in articles of the same type, and in articles with which we compared our results. Third, we consulted an experienced statistician, and her opinion was that the Bonferroni correction would not be necessary in this case. Forth, several sources consider this correction to be overly conservative and to increase the chance of type II errors. Finally, this study also does not reach final definite conclusions and serves as a starting point to more studies, as is pointed out in the discussion. With this in mind, we find it acceptable not to use Bonferroni corrected values, as being overly conservative could eliminate some of the correlations we have found in this study.

We also provide some links where the reasons not to make this correction is explained:

GraphPad Prism 10 Statistics Guide - When it makes sense to not correct for multiple comparisons No adjustments are needed for multiple comparisons - PubMed

No adjustments are needed for multiple comparisons - PubMed.

Adjusting for multiple testing—when and how? - ScienceDirect

Reviewer 2 Report

Comments and Suggestions for Authors

This was a retrospective study that aimed to investigat the changes in placentas from obese women with and without SARS-CoV-2. To achieve this, a cohort of women were identified between 2017 and 2023. These were divided into four groups non-obese non-SARs-CoV-2 infected (n=17) and infected (N=19), obese non-SARs-CoV-2 infected (N=14) and SARs-CoV-2 infected (N=12). To control for gestational diabetes (GDM), they included women with GDM in the obese groups. The key findings were an increased risk of placental lesions related to both maternal and fetal malperfusion and ischemic injury to the placenta in those infected with SARs-CoV-2.

The findings in this study would be of interest but the paper is very difficult to follow and would require significant revisions. Furthermore, the authors confuse the study by included 2 groups of women - non-obese meaning the study was now comparing placentas from 4 groups. It would have been easier to follow if they focused only on 2 groups. By extending the study in the pre-COVID period they introduce another variable - time in the study.  If the study aimed to compare placenta pathology in 4 groups then the title must be revised as well as the objectives. The manuscript must be shortened and to the point.

It would be better to have a structured abstract and also present the manuscript as standard - i.e. results and then discussions. By combining the results and discussion, the paper becomes very difficult to follow. The abstract is also difficult to follow just because of the way its structured.

Specific comments

  1. leave out the word prevenient - it adds nothing to the sentences where its used (abstract -line 19 etc)
  2. There should be numbers in the abstract rather than just being narrative- Structuring it would bring this out.
  3. The last sentence of the abstract is incomplete. Revise it please
  4. See comments above about period of study etc. It would have been better to just examine placenta obtained at the time of COVID. Unless I can be assured that all placentas are examined in this hospital, there would be suspicion that the placentas examined prior to COVID were for other reasons that might have affected the pathology of the placentas. There could be justification in examining all placentas during COVID but then was this justifiable since this is a retrospective study?
  5. Line 101 - it seems those from who SARs-CoV-2 was recovered at the time of delivery were excluded and those with positive PCR included. What is the difference, or I am missing something? Please clarify.
  6. Why were GDM's included? When was GDM diagnosed in these women? If you did the analysis without the GDM, how different would the results be? I think you have watered your findings by including a variable that may have pathology that varies based on the timing of the GDM.
  7. Not sure about the relevance of most of the information at. lines 127-129. For example, previous abortions, IVF, mode of delivery, APGAR score, gender of the baby, weight etc. Surely co-morbidities were an exclusion criterion - how then did you collect these?
  8. Tables are too busy and could be simplified to help the reader. Format Table 1 - the age of the OB+SARS is off-line. Also be consistent with your decimals. In the narrative at lines 161-162 there is on decimal but there are 2 in the Table confusing the reader (e.g. 32.37 and 32.4). Simplify the Table by leaving out mode of delivery, APGAR scores, gender and birth weight as these add nothing to the data. Could the columns on p-values be removed and where there is significance shown as asterisk etc? Thid would make the tables less busy.
  9. Line 257 - what does 'statistically similar' mean?
  10. Throughout parts of the narrative, instead of citing references properly, you simply state numbers. This is incorrect - Lines 267,269, 273 and indeed the section from lines 267-282. Just one example at line 267 you state "Finally [57] also found... It should be Finally Brien et al [57} also found.... Please correct.
  11. Line 307 - change verified to shown
  12. At lines 355-357 you seem to imply that had your numbers increased your findings would have been statistically significant. You cannot say this. All you can say is that the results might have been different.
  13. Lines 367-368 - I do not agree with the statement that you controlled for GDM. You simply included cases of GDM which is a very broad category of diabetes in pregnancy. You should leave this group out.
  14. The last sentence is too long and should be revised and shortened to make it clearer.
Comments on the Quality of English Language

The language will need some revising to make it easier to follow.

Author Response

Response to Reviewer 2 Comments

Thank you very much for taking the time to review this manuscript. Please find the detailed responses below and the corresponding revisions/corrections highlighted/in track changes in the re-submitted file.

1. The English could be improved to more clearly express the research.

Response 1. The manuscript has been reviewed for English improvement.

2. Questions for General Evaluation

Reviewer’s Evaluation

Response and Revisions

Does the introduction provide sufficient background and include all relevant references?

Can be improved

The Introduction was changed (lines 86-91)

Is the research design appropriate?

Must be improved

Assessed in response 6, 7 and 8

Are the methods adequately described?

Must be improved

Assessed in response 9 and 10

Are the results clearly presented?

Must be improved

Sub-sections were added to the Results and Discussion section

Are the conclusions supported by the results?

Are all figures and tables clear and well-presented?

Can be improved

Must be improved

The Abstract was changed, the Conclusions were revised

Some Tables were changed in order to improve them

3. Point-by-point response to Comments and Suggestions for Authors

Comment 1: The findings in this study would be of interest but the paper is very difficult to follow and would require significant revisions.

Response: We hope that after revision the paper is easier to follow.

Furthermore, the authors confuse the study by included 2 groups of women - non-obese meaning the study was now comparing placentas from 4 groups. It would have been easier to follow if they focused only on 2 groups.

Response: Yes, we have 4 groups of placentas because we have two variables: obese/non-obese and Covid/no-Covid. These 4 groups are mandatory. If we had only two groups (obese with Covid/no-Covid) we could not conclude if the changes induced by Covid are similar in obese and non-obese. If we had only two groups (Obese or non-obese women with Covid), we could not eliminate possible changes induced by obesity alone. So, the OB and the Control were added to control for possible confounders, as there are placental lesions that occur even when there is no comorbidities/disease in the pregnant woman, and there are lesions that are more common to be observed in obese patients than in non-obese.

By extending the study in the pre-COVID period they introduce another variable - time in the study.

Response: The only reason for having included a group of patients deviated in time from the others (OB) relates to the limitations in obtaining new histological reports of OB patients. So, we decided to include samples that already had the pathology report available in this study. Anyway, we do not think that differences in placental pathological changes/pathology report are expected to exist in such a short time lag (2017-2023). Of note, the reports were made by the same pathologist, irrespective of the year.

If the study aimed to compare placenta pathology in 4 groups then the title must be revised as well as the objectives. The manuscript must be shortened and to the point.

Response: in agreement with your suggestion, the title was revised, in order to more clearly show that both the influence of Sars-Cov-2 infection and maternal obesity were assessed. The Aims were also changed in the Abstract (lines 20-21) and Introduction (lines 89-91).

Comment 2: It would be better to have a structured abstract and also present the manuscript as standard - i.e. results and then discussions. By combining the results and discussion, the paper becomes very difficult to follow. The abstract is also difficult to follow just because of the way its structured.

Response 2: In relation to the Abstract, it is now structured as requested. Anyway, we made changes in the Abstract text, in order to make it clearer. In relation to the fact that the Results and Discussion are combined, we decided to join them exactly because we thought that in that way, it would be easier to follow the manuscript. It would also reduce the size of the manuscript. Anyway, sub-sections were added to this section to make it easier to follow. Moreover, we made several changes in the text that we think will improve the clarity of the manuscript, but if you still think we must separate Results and Discussion, we will do it (although it will increase the manuscript size).

Comment 3: Leave out the word prevenient - it adds nothing to the sentences where its used (abstract -line 19 etc)

Response 3: Thank you for this comment, the issue has been corrected (in Abstract and Methods-line 98).

Comment 4: There should be numbers in the abstract rather than just being narrative- Structuring it would bring this out.

Response 4: Thank you for pointing this out, to resolve this we did introduce abstract structure headings and we introduced some numbers (lines 26 and 28). Moreover, the Abstract text itself was changed. We think the Abstract was improved.

Comment 5:    The last sentence of the abstract is incomplete. Revise it please.

Response 5: Thank you for this comment, the last sentences of the abstract were changed (lines 29-32).

Comments6: See comments above about period of study etc. It would have been better to just examine placenta obtained at the time of COVID. Unless I can be assured that all placentas are examined in this hospital, there would be suspicion that the placentas examined prior to COVID were for other reasons that might have affected the pathology of the placentas. There could be justification in examining all placentas during COVID but then was this justifiable since this is a retrospective study?

Response 6: Because most of the OB and OB+SARS women also present other pathologies (pregnancy or non-pregnancy related), it is difficult to obtain placental samples from obese patients having no other pathology. That is why we decided to include these ‘historic’ samples, that already had the pathology report available for us to include in this study. These samples were obtained from hospital records and selected to include obese mothers without any other relevant pathology. This is the only reason a group of this nature (deviated in time from the others) was included. Moreover, because no pathological examination of the placenta is made in the hospital when the mothers belong to the Control group, we had to collect these, and that is why these samples are deviated in time from the others. Nevertheless, all the samples were analyzed in the same hospital by the same pathologist.

Comment 7: Line 101 - it seems those from who SARs-CoV-2 was recovered at the time of delivery were excluded and those with positive PCR included. What is the difference, or I am missing something? Please clarify.

Response 7: The difference is an active infection at the time of delivery. Our study only included SARS-CoV-2 active infection at the time of delivery (PCR positive). So, if the pregnant person tested negative at delivery, even if she had this infection during pregnancy, she was excluded. With this we eliminated differences that might arise from differences in the time of infection.

Comment 8: Why were GDM's included? When was GDM diagnosed in these women? If you did the analysis without the GDM, how different would the results be? I think you have watered your findings by including a variable that may have pathology that varies based on the timing of the GDM.

Response 8: Thank you for this comment. We included GDM participants because, without them, the OB and the OB+SARS samples size would be even smaller. Indeed, it is very uncommon that an obese pregnant woman is not diagnosed with GDM, and so it would be very difficult to perform a study with obese pregnant patients excluding GDM cases. Anyway, we were aware of this point, and so we added a similar % of participants with GDM in all groups (lines 123-129). In relation to the time of GDM diagnosis, screening and diagnosis of GDM in Portugal is always made at mid-pregnancy (24-28 weeks of gestation).

Comment 9: Not sure about the relevance of most of the information at. lines 127-129. For example, previous abortions, IVF, mode of delivery, APGAR score, gender of the baby, weight etc. Surely co-morbidities were an exclusion criterion - how then did you collect these?

Response 9: We agree with this comment, and in line with it, previous abortions and IVF were deleted from the data collected and the gender of the newborn was removed from Table 1 (see also lines 131-133). However, we did not delete the information concerning the presence or not of GDM and HT, because we think these two co-morbidities are important in the context of the present study (comparison between obese and non-obese women). Also, we did not delete the newborn weight and APGAR score because we believe it to be an important parameter in the context of fetal programming (the newborn weight has programming effects). Similarly, the mode of delivery may have consequences, as cesarean section increases the risk of perinatal mortality.

Comment 10: Tables are too busy and could be simplified to help the reader. Format Table 1 - the age of the OB+SARS is off-line. Also be consistent with your decimals. In the narrative at lines 161-162 there is on decimal but there are 2 in the Table confusing the reader (e.g. 32.37 and 32.4).

Simplify the Table by leaving out mode of delivery, APGAR scores, gender and birth weight as these add nothing to the data.

Could the columns on p-values be removed and where there is significance shown as asterisk etc? Thid would make the tables less busy.

Response 10: We agree with this comment, and Table 1 was re-formatted, and the decimals were uniformized (all changes are marked in yellow). Moreover, in line with the previous comment and response, we eliminated the gender of the newborn from Table 1 (see also lines 131-133). However, we did not delete the newborn weight and APGAR score because we believe it to be an important parameter in the context of fetal programming (the newborn weight has programming effects). Similarly, the mode of delivery may have consequences, as cesarean section increases the risk of perinatal mortality. Finally, the p -values were replaced by “*”.

Comment 11: Line 257 - what does 'statistically similar' mean?

Response 11: Thank you for pointing this out. It is a redundant statement, and has been deleted (line 263).

Comment 12: Throughout parts of the narrative, instead of citing references properly, you simply state numbers. This is incorrect - Lines 267,269, 273 and indeed the section from lines 267-282. Just one example at line 267 you state "Finally [57] also found... It should be Finally Brien et al [57} also found.... Please correct.

Response 12: Thank you for pointing this out. This issue has been corrected in lines 273, 275, 281, 282, 286, 287, 290, 315, 338, 357 and 358.

Comment 13: Line 307 - change verified to shown

Response 13: Thank you for this comment, the word was replaced (line 315).

Comment 14: At lines 355-357 you seem to imply that had your numbers increased your findings would have been statistically significant. You cannot say this. All you can say is that the results might have been different.

Response 14: Thank you for pointing this out. Yes, we cannot imply this. The text is updated in the manuscript (lines 377-379).

Comment 15: Lines 367-368 - I do not agree with the statement that you controlled for GDM. You simply included cases of GDM which is a very broad category of diabetes in pregnancy. You should leave this group out.

Response 15: We included GDM participants because, without them, the OB and the OB+SARS sample sizes would be even smaller. Indeed, it is very uncommon that an obese pregnant woman is not diagnosed with GDM, and so it would be very difficult to perform a study with obese pregnant patients excluding GDM cases. Anyway, we were aware of this point, and so we added a similar % of participants with GDM in all groups (lines 123-129). This way, the possible influence of GDM in the placental pathology is expected to be similar in all groups.

Comment 16: The last sentence is too long and should be revised and shortened to make it clearer.

Response 16: The last sentence has been shortened (lines 387-389).

Reviewer 3 Report

Comments and Suggestions for Authors

The article presents both original and field-relevant contributions. Here’s an analysis of its originality and how it addresses a specific gap in the field. The study fills a specific research gap by exploring how maternal obesity combined with active SARS-CoV-2 infection affects placental histopathology.

  1. The study acknowledges small group sizes (n=12). Increase sample size and include a power analysis to support statistical conclusions and reduce the risk of type II errors.
  2. GDM and hypertension were not excluded due to small group sizes, yet they are known confounders in placental pathology.
  3. Several data points (infection severity, some placental evaluations) were missing. Apply imputation techniques or sensitivity analyses and improve data completeness in future work.

In summary, while the study is methodologically solid with its prospective, controlled, and randomized design, integrating these improvements would significantly enhance the internal and external validity, reduce bias, and provide stronger evidence for clinical recommendations.

Author Response

Response to Reviewer 3 Comments

1. Questions for General Evaluation

Reviewer’s Evaluation

Response and Revisions

Does the introduction provide sufficient background and include all relevant references?

Can be improved

The Introduction was changed (lines 86-91)

Is the research design appropriate?

Yes

Are the methods adequately described?

Yes

Are the results clearly presented?

Yes

Are the conclusions supported by the results?

Are all figures and tables clear and well-presented?

Yes

Yes

2. Point-by-point response to Comments and Suggestions for Authors

Comment 1: The study acknowledges small group sizes (n=12). Increase sample size and include a power analysis to support statistical conclusions and reduce the risk of type II errors.

Response 1: Thank you for pointing this out. Unfortunately, it is not possible to increase the numbers of patients included in this analysis. In the case of SARS and OB+SARS groups, screening of COVID-19 is currently not done in the hospital, so it is impossible to increase the number of patients in these groups. But we acknowledge that studies with more patients included are more statistically solid.

Comment 2: GDM and hypertension were not excluded due to small group sizes, yet they are known confounders in placental pathology.

Response 2: We included GDM participants because, without them, the OB and the OB+SARS sample sizes would be even smaller. Indeed, it is very uncommon that an obese pregnant woman is not also diagnosed with GDM, and so it would be very difficult to perform a study with obese pregnant patients excluding GDM cases. We were aware of this potential confounding factor, so we included a similar percentage of participants with GDM in all groups (lines 123-129). This approach is expected to minimize the influence of GDM on placental pathology across the groups.

Comment 3: Several data points (infection severity, some placental evaluations) were missing. Apply imputation techniques or sensitivity analyses and improve data completeness in future work.

Response 3: Yes, some information is missing, but this was a retrospective study, and some placental pathology records or information collected by the patient’s doctors were incomplete in the consultations reports and medical records. So, it was impossible for us to obtain the complete information from the records. But we agree that this is a limitation of this study, and this is mentioned (lines 376-378).

Reviewer 4 Report

Comments and Suggestions for Authors Abstract
The abstract is the reader’s first point of contact with the manuscript; therefore, it must be prepared with great care. I have the following suggestions:
  1. Briefly articulate why it is important to investigate the combined effect of SARS-CoV-2 infection and obesity on the placenta.
  2. It would be useful to indicate that the control and OB (obese) groups are from earlier years, whereas the SARS and OB+SARS groups are from later years. This temporal difference may introduce bias in the comparisons.
  3. The conclusions presented are strong (e.g., “increased risk”), but there is no reference to effect size or clinical significance. For at least one of the claims, it would be important to indicate the magnitude of the observed differences. I understand that authors often avoid presenting specific data in the abstract to encourage full-article downloads, but in this case, including some data would be advisable.
Introduction
4) The introduction is overall well-structured, but more emphasis could be placed on the specific vulnerability of the placenta to the dual impact of obesity and infection. This would help the reader appreciate the scientific significance of the study.
5) While I acknowledge the importance of using recent references—and I also tend to prioritize them—the sample collection ended in 2023, implying that the study design predates this. Therefore, I suggest including older references in the Introduction to frame the original rationale, while using more recent literature in the Discussion to support the relevance of the research. This approach avoids the impression that the study was retrospectively justified.
6) The aim presented in the final paragraph is clear and relevant but somewhat convoluted in its current form. My suggested revision is:
“Given the limited data on how the coexistence of SARS-CoV-2 infection and maternal obesity affects placental pathology, this study aimed to compare placental histopathological changes in obese versus non-obese mothers with active SARS-CoV-2 infection.”   Methods
7) As I have mentioned previously, this represents a fundamental flaw in the study. Although the four groups are clearly defined, the historical nature of the OB group (2017–2019) presents a serious risk of bias due to potential differences in diagnostic protocols, placental examination practices, or technical standards. This could have been mitigated by collecting control and OB samples during the COVID-19 pandemic, allowing for comparisons both before and during the pandemic. While this cannot be remedied now, the limitations should be acknowledged explicitly.
8) The partial inclusion of GDM and hypertension is understandable, but these are potential confounding factors, especially in the OB and OB+SARS groups. Therefore, it is recommended to account for them rigorously in the statistical analysis (e.g., through covariate models).   Results and Discussion
9) A general comment: this section lacks structure and reads as a single block of text. Dividing it into subsections would significantly improve clarity and readability.
10) Including the causes of preterm birth is commendable, but it would also be useful to note that preterm birth can bias other parameters, such as placental weight or neonatal weight.
11) Macroscopic examination of the placenta: The main finding—that there were no differences in macroscopic parameters—is important. However, was a difference expected in the first place? A brief discussion would be helpful to determine whether these findings support or contradict previous studies.
12) The reported odds ratio for hypoxic-ischemic lesions (e.g., a 15-fold increase in OB+SARS vs. control) is noteworthy. However, a more in-depth physiological explanation would further strengthen the argument.
13) Including a well-composed histological image (e.g., H&E staining showing subchorionic fibrin deposition or an example of MVM) would greatly enhance clarity for the reader and improve the overall quality of the publication. I strongly recommend adding such an illustration.
14) Based on my experience, the link between obesity, overweight status, and the severity of associated conditions is a highly relevant topic. Emphasizing this connection would increase the impact of the study. This is not mandatory, but a strong recommendation.     15)
English is not my native language, yet I am convinced that the manuscript requires language revision. I recommend a native speaker or professional translator review it. The text is often difficult to read and interpret, with overly dense sentences and the use of inappropriate expressions. While I typically do not struggle with English academic texts, I found this one challenging at times.  

Author Response

Response to Reviewer 4 Comments

1. Questions for General Evaluation

Reviewer’s Evaluation

Response and Revisions

Does the introduction provide sufficient background and include all relevant references?

Yes

Yes

Is the research design appropriate?

Yes

Are the methods adequately described?

Yes

Are the results clearly presented?

Yes

Are the conclusions supported by the results?

Yes

2. Point-by-point response to Comments and Suggestions for Authors

Abstract

Comment 1: Briefly articulate why it is important to investigate the combined effect of SARS-CoV-2 infection and obesity on the placenta.

Response 1: This information was added to the Abstract (lines 17-19).

Comment 2: It would be useful to indicate that the control and OB (obese) groups are from earlier years, whereas the SARS and OB+SARS groups are from later years. This temporal difference may introduce bias in the comparisons.

Response 2: We agree with this comment, and have modified the abstract (line 25).

Comment 3: The conclusions presented are strong (e.g., “increased risk”), but there is no reference to effect size or clinical significance. For at least one of the claims, it would be important to indicate the magnitude of the observed differences. I understand that authors often avoid presenting specific data in the abstract to encourage full-article downloads, but in this case, including some data would be advisable.

Response 3: We agree with this comment, but due to restrictions in total word count for the Abstract (up to 200 words), we were forced to overly reduce the amount of information presented, and summarize as much as possible. Anyway, we were able to introduce some magnitude numbers (lines 26-28).

Introduction

Comment 4: The introduction is overall well-structured, but more emphasis could be placed on the specific vulnerability of the placenta to the dual impact of obesity and infection. This would help the reader appreciate the scientific significance of the study.

Response 4: Thank you for this comment. We added the requested information (lines 86-89).

Comment 5:    While I acknowledge the importance of using recent references—and I also tend to prioritize them—the sample collection ended in 2023, implying that the study design predates this. Therefore, I suggest including older references in the Introduction to frame the original rationale, while using more recent literature in the Discussion to support the relevance of the research. This approach avoids the impression that the study was retrospectively justified.

Response 5: In response to this, we updated citations with publications prior to 2023 in the Introduction, by introducing older references. The new citations are 11-13 (lines 63), 18-20 (line 74). These are also highlighted in the reference list.

Comment 6: The aim presented in the final paragraph is clear and relevant but somewhat convoluted in its current form. My suggested revision is:

“Given the limited data on how the coexistence of SARS-CoV-2 infection and maternal obesity affects placental pathology, this study aimed to compare placental histopathological changes in obese versus non-obese mothers with active SARS-CoV-2 infection.”

Response 6: Thank you for this comment. This has been previously mentioned as well by another reviewer, and the sentence was changed accordingly (lines 89-91).

Methods

Comment 7: As I have mentioned previously, this represents a fundamental flaw in the study. Although the four groups are clearly defined, the historical nature of the OB group (2017–2019) presents a serious risk of bias due to potential differences in diagnostic protocols, placental examination practices, or technical standards. This could have been mitigated by collecting control and OB samples during the COVID-19 pandemic, allowing for comparisons both before and during the pandemic. While this cannot be remedied now, the limitations should be acknowledged explicitly.

Response 7: Based on limitations of obtaining new reports of obese only patients, we decided to include these ‘historic’ samples, that already had the pathology report available for us to include in this study. This is the only reason a group of this nature (deviated in time from the others) was included. Regarding your concerns towards the potential differences in diagnostic protocols, placental examination practices, or technical standards, we can assure you that these samples were analyzed in the same hospital by the same pathologist.

Comment 8: The partial inclusion of GDM and hypertension is understandable, but these are potential confounding factors, especially in the OB and OB+SARS groups. Therefore, it is recommended to account for them rigorously in the statistical analysis (e.g., through covariate models).

Response 8: We included GDM participants because, without them, the OB and the OB+SARS sample sizes would be even smaller. Indeed, it is very uncommon that an obese pregnant woman is not diagnosed with GDM, and so it would be very difficult to perform a study with obese pregnant patients excluding GDM cases. Anyway, we were aware of this point, and so we added a similar % of participants with GDM in all groups (lines 123-129). This way, the possible influence of GDM in the placental pathology is expected to be similar in all groups.

Results and Discussion

Comment 9: A general comment: this section lacks structure and reads as a single block of text. Dividing it into subsections would significantly improve clarity and readability.

Response 9: In relation to the fact that the Results and Discussion are combined, we decided to join them exactly because we thought that in that way, it would be easier to follow the manuscript. It would also reduce the size of the manuscript. We made several changes in the text that we think will improve the clarity of the manuscript (including sub-headings), but if you still think we must separate Results and Discussion, we will do it (although it will increase the manuscript size).

Comment 10: Including the causes of preterm birth is commendable, but it would also be useful to note that preterm birth can bias other parameters, such as placental weight or neonatal weight.

Response 10: We agree with this comment, and a sentence concerning this point was included in the manuscript (lines 174-176).

Comment 11: Macroscopic examination of the placenta: The main finding—that there were no differences in macroscopic parameters—is important. However, was a difference expected in the first place? A brief discussion would be helpful to determine whether these findings support or contradict previous studies.

Response 11: Thank you for pointing this out, we have provided a more detailed discussion of this matter (lines 216-219). But it is important to note that most of the previous studies focused on microscopic findings in the placentas, and so we do not have a lot of results to compare to.

Comment 12: The reported odds ratio for hypoxic-ischemic lesions (e.g., a 15-fold increase in OB+SARS vs. control) is noteworthy. However, a more in-depth physiological explanation would further strengthen the argument.

Response 12: We agree with this comment and have added a more detailed description of the nature and implications of these lesions. (lines 298-302).

Comment 13: Including a well-composed histological image (e.g., H&E staining showing subchorionic fibrin deposition or an example of MVM) would greatly enhance clarity for the reader and improve the overall quality of the publication. I strongly recommend adding such an illustration.

Response 13: Thank you for this comment. In order to obtain the histological images, we would need more time to contact the pathologist team that analyzed these placentas. A few months, at least.

Comment 14: Based on my experience, the link between obesity, overweight status, and the severity of associated conditions is a highly relevant topic. Emphasizing this connection would increase the impact of the study. This is not mandatory, but a strong recommendation.

Response 14: We already mention this link in the original manuscript (in the last 3 paragraphs of the Introduction). Moreover, the last paragraph of the Introduction was changed, and reference to the link between obesity and SARS-Cov-2 at the placental level was added (lines 86-89).

Comment 15: English is not my native language, yet I am convinced that the manuscript requires language revision. I recommend a native speaker or professional translator review it. The text is often difficult to read and interpret, with overly dense sentences and the use of inappropriate expressions. While I typically do not struggle with English academic texts, I found this one challenging at times.  

Response 15: Thank you for this comment, as per suggestion not only of you but also of some other reviewers, the manuscript has been reviewed by the authors for English improvement.

Round 2

Reviewer 1 Report

Comments and Suggestions for Authors

I had the oportunity ti review the manuscript "Placental pathology of SARS-CoV-2 affected pregnancies of 2 obese and non-obese patients"

Tha authors improved the revised manuscript.

However, the main criticism is still the same. The sample size is small and it results in probable overestimation of the findings. The authors claim that theirs is an exploratory analysis, and that's the reason they do not apply methods to reduce overestimation of statistical significances. 

The authors should state this (the exploratory character of the study) as one of the limitations of the study.

Tables should be improved. In their present form, Tables 1, 3, and 4 are half empty, because of the columns devised for p values. The authors should find another way to display p values that do not result in empty cells.

Text should be revised as well. For instances, the sentence "they were significantly more present" should read "they were significantly more frequent"

Author Response

Response to Reviewer #1 Comments

Thank you very much for taking the time to review this manuscript. Please find the detailed responses below and the corresponding revisions/corrections highlighted in the re-submitted file.

Comment 1: However, the main criticism is still the same. The sample size is small and it results in probable overestimation of the findings. The authors claim that theirs is an exploratory analysis, and that's the reason they do not apply methods to reduce overestimation of statistical significances. 

The authors should state this (the exploratory character of the study) as one of the limitations of the study.

Response 1: In view of this comment, we added some text in the limitations of the study (see lines 390-391).

Comment 2: Tables should be improved. In their present form, Tables 1, 3, and 4 are half empty, because of the columns devised for p values. The authors should find another way to display p values that do not result in empty cells.

Response 2: Thank you for addressing this issue. We have changed the style of these tables, so that the empty space is reduced. Since there were no significant comparisons in table 3, we eliminated the cells that were meant to show comparisons between groups and added the information in the legend of the table. Please see Tables 1, 3 and 4.

Comment 3: Text should be revised as well. For instances, the sentence "they were significantly more present" should read "they were significantly more frequent"

Response 3: Thank you for addressing this issue, we have corrected the text in the manuscript (see line 325).

Reviewer 2 Report

Comments and Suggestions for Authors

Thank you for making an effort to address my comments. I am happy with your revisions.

Author Response

Thank you very much.

Reviewer 4 Report

Comments and Suggestions for Authors

The authors have provided acceptable responses to my comments and questions. However, the text requires revision. For example, et al is consistently used incorrectly (the correct form is et al. with a period). Additionally, the list of abbreviations should be arranged in alphabetical order.

Author Response

Response to Reviewer #4 Comments

Thank you very much for taking the time to review this manuscript. Please find the detailed responses below and the corresponding revisions/corrections highlighted in the re-submitted file.

Comment 1: For example, et al is consistently used incorrectly (the correct form is et al. with a period).

Response 1: Thank you for addressing this issue, we have corrected this throughout the manuscript.

Comment 2: Additionally, the list of abbreviations should be arranged in alphabetical order.

Response 2: We have updated the order of this list; it is now in alphabetical order.